# Designing of immunodiagnostic assay using polyclonal antibodies for detection of Enteropathogenic *Escherichia coli* strains

**Aliyi Hassen Jarso**[1,2], **Biniam Moges Eskeziyaw**[3], **Degisew Yinur Mengistu**[4], **Tesfaye Sisay Tessema**[5]*

1 Center for Innovative for Drug Development and Therapeutic Trials for Africa (CDT-Africa), Addis Ababa University, Addis Ababa, Ethiopia, 2 Department of Biology, Ambo University, Ambo, Ethiopia, 3 Department of Biotechnology, Debre Berhan University, Debre Berhan, Ethiopia, 4 Institute of Biotechnology, University of Gondar, Gondar, Ethiopia, 5 Institute of Biotechnology, Addis Ababa University, Addis Ababa, Ethiopia

* tesfaye.sisayt@aau.edu.et

**Data Availability Statement:** All relevant data are within the paper.

## Abstract

Enteropathogenic *Escherichia coli* (EPEC) is a significant bacterial pathogen that causes infantile diarrhea, particularly in low- and middle-income countries. The lack of a reliable diagnostic method greatly contributes to the increased occurrence and severity of the disease. This study aimed at developing of a cost-effective, rapid, and efficient immunodiagnostic assay for detecting EPEC infection. Lipopolysaccharide (LPS) was extracted from overnight EPEC cultures and combined with alum adjuvant, and then injected into mice for three rounds of immunizations. Subsequently, sera were collected after each immunization and utilized in agglutination assays conducted on glass slides. Both the LPS and colonies of the EPEC isolate used for LPS preparation were employed in these agglutination assays. To evaluate the assay's performance, a total of 34 bacteria, which comprise pathogenic, non-diarrheic *E. coli* and non-*E. coli* pathogenic bacteria were used. The developed assay detected EPEC, which yielded positive reactions within 6 minutes on average for both purified LPS and bacterial isolates. The assay exhibited 100% sensitivity and a 95.83% specificity for the detection of EPEC local isolates. Moreover, the assay also detected a low number of bacteria forming units ($104 \times 10^4$ CFU/ml) in spiked fecal samples. This study conclusively confirms that the developed immunodiagnostic assay possesses multiple favorable characteristics, including user-friendliness, high sensitivity, high specificity, cost-effectiveness, and time-efficiency. Hence, this assay can be used as ideal diagnostic assay, which is highly suitable for the detection and screening of EPEC infection in both humans and cattle in one health perspective of resource-limited laboratories.

## Introduction

Enteropathogenic *Escherichia coli* (EPEC) is one of the diarrhea-causing E. coli pathotypes that was first described as the most important pathogen for infecting infants [1]. It causes persistent diarrhea leading to death in children under 5 years of age worldwide and is prevalent in

**Funding:** This study was supported by the Bio and Emerging Technology Institute (BETin), Prof Tesfaye Sisay Tessema won research grant. The funders had no role in study design, data collection and analysis, decision to publish, or preparation of the manuscript.

**Competing interests:** no competing interest

both community and hospital settings [2, 3] EPEC is associated with rare outbreaks in developed countries and most outbreaks are typically recorded in nurseries and daycare centers [4, 5]. Community-acquired EPEC infection rates were previously thought to be highest in the first six months after birth; however, subsequent studies in children up to 5 years of age suggest that prevalence increases with age [6].

Infection with EPEC mainly occurs through direct person-to-person contact and through ingestion of contaminated food. Typical symptoms of EPEC infection include severe vomiting, watery diarrhea, mild fever and dehydration. In addition, EPEC infection can lead to severe malabsorption of nutrients, lactose intolerance and food allergies, further exacerbating nutritional deficiencies and prolonging diarrhea [7]. A key feature of EPEC pathogenicity is the formation of attaching and effacing (A/E) lesions, which are mediated by a chromosomal pathogenicity island, the locus of enterocyte effacement (LEE). In addition, the LEE region of EPEC encodes important proteins such as a type III secretion system, lytic transglycosylase (EtgA), molecular chaperones, outer membrane protein intimin, and transcriptional regulators (Ler, GrlR, and GrlA) [8, 9]

Depending on the presence of the EPEC adherence factor plasmid (pEAF), there are two types of EPEC, the typical EPEC (tEPEC) and the atypical EPEC (aEPEC) [10].The typical EPEC (tEPEC) are mainly characterized by *eae*+ *bfpA*+, while atypical EPEC (aEPEC) strains are characterized by *eae*+ *bfpA*−, which do not comprise EAF plasmid and are defined as negative for bfpA [10]. Atypical EPEC (aEPEC) is more common than typical EPEC (tEPEC) in children under five years of age with diarrhea and is emerging as a drug-resistant enteropathogen for public care [3]. However, the importance of EPEC as a diarrheal pathogen is underestimated because efficient and rapid diagnostic methods are not available in routine healthcare laboratories. This argues for the development of drug-resistant typical EPEC and leads to frequent outbreaks of typical EPEC in children. Identifying the type of pathogenic *E. coli* causing the diarrhea during diagnosis plays an important role in the treatment and transmission of the disease. To this end, the development of an effective and rapid detection of this pathogen is crucial to alleviate the spread of the pathogen and threatening outbreaks.

Various approaches have been developed to detect EPEC, such as immunological (serotypic), phenotypic and genotypic methods. Bacterial cultures and fluorescence microscopy are used for phenotypic detection, while DNA hybridization or polymerase reaction methods are used for genotypic detection [3]. Genotypic tests such as DNA probe hybridization or PCR targeting EAF, *eae* and *bfpA* are preferred for identification of EPEC. Despite these advantages, molecular genotypic tests have disadvantages due to allelic variability and the need for specific primers. For example, the virulence gene can be identified by PCR; however this gives no indication of virulence factor expression. In addition, the genetic diversity of EPEC strains and the need for a well-equipped laboratory with sophisticated instruments and qualified personnel hinder the advantage of molecular detection methods [11].

Immunological methods, such as enzyme immunoassays, have been used to detect pathogenic *E. coli*. However, performing these ELISAs can be laborious and time-consuming [12]. In addition, enzyme immunoassays are expensive and complicated to use, making them inaccessible in resource-limited settings to meet public health needs [13]. Compared to other immunological assays, slide agglutination test is a suitable serological diagnostic technique as it is rapid, simple to perform and cheap in poor laboratory settings [14]. Such assays are not easily available for the underdeveloped world. Moreover, most of the available diagnostics focus on Shiga toxin like producing *E. coli* (STEC) strains and no assay has been published for detection of EPEC strains. EPEC was identified based on the O:H serotype, but this is no longer necessary for a strain to be considered EPEC since multiple *E. coli* O serotypes are closely associated with each other [11, 15]. Most EPEC strains adhere to epithelial cells in vitro, which

is detectable by the fluorescent actin staining (FAS) test, although many laboratories do not have the necessary tissue cultures for such tests [11, 16].

The development of a diagnostic test that overcomes the above-mentioned disadvantages is crucial for effective treatment and a short diagnosis time of EPEC disease. The most important and common virulence factors such as intimin, a secreted type III protein, the bundle-forming pilus (BFP) and several activators of host pattern recognition receptors, including lipopolysaccharide and flagellin, serve to recognize typical EPEC and atypical EPEC infections [17–19]. In this study, we produced polyclonal antibodies from lipopolysaccharide extracts of locally isolated EPEC strains and developed a rapid, simple, efficient and cost-effective immunodiagnostic assay for the detection of EPEC.

## Materials and methods

### Bacterial strains selection and growth conditions

In this study 30, *E. coli* isolates (10 STEC strains, 10 EPEC strains, *10 E. coli* isolates from non-diarrheic calves) were obtained from Prof. Tesfaye Sisay's laboratory, Institute of Biotechnology, Addis Ababa University. Additionally, the *non E. coli* bacterial spp (*Shigella flexinari*, *Salmonella Typhimurium*, *Staphylococcus aureus*, and *Klebsiella pneumoniae*) were obtained from Ethiopian public health institute (EPHI). These *E. coli* strain isolates were isolated originally from various parts of Ethiopia and were confirmed and identified by PCR assays using specific primers targeting the intimin (*eae*), bundle forming pili (*bfp*) and shiga toxin (*stx*) genes and stored as glycerol stocks at– 20°C in the laboratory [20, 21].

The EPEC (k51) isolate used for LPS extraction was cultured on Eosin Methylene Blue (EMB) agar overnight at 37°C and re-cultured into tryptose soya agar (TSA). Single colony was inoculated into 300 ml of tryptose soya broth (TSB), divided into different test-tubes and incubated for 24 hrs at 37°C. Then from each test tube 1ml of culture was used to determine the bacterial optical density by spectrophotometer at 600 nm.

### Enteropathogenic *Escherichia coli* (EPEC k51) Lipopolysaccharide (LPS) antigen extraction and purification

Lipopolysaccharide (LPS) was extracted from the EPEC k51 strain using Chloroform-methanol method [22] with some modification as follows. About 298 ml of the overnight grown broth culture were centrifuged at 6000 rpm at 4°C for 30 minutes. The supernatant was discarded and the pellet was re-suspended in 2 ml of 96% ethanol alcohol by vortexing and then centrifuged at 10,000 rpm for 10 min and this step was repeated four times. Finally, the pellet was collected and dried inside the hood to completely evaporate the alcohol. The dried pellet was transferred into 2 ml Eppendorf tube and re-suspended in 1ml of 10% EDTA and then incubated on ice for 15 min. Approximately 1ml of saturated methanol/chloroform (333 μl:666 μl ratios) was added to the bacterium-EDTA solution and kept on shaker for two hours following 10 min centrifugation at 10,000 rpm. From top to bottom three layers including methanol, left biomass including cell lysate and chloroform layer were formed. The biomass layer with cell lysate was discarded and chloroform and methanol layers were separated and poured into 50 ml glass beaker to permit complete and quick evaporation of methanol and chloroform layers. The purified LPS (dried pellet) was dissolved with sterilized PBS and stored at—20°C until further used.

### Evaluation of the purity of the purified EPEC LPS antigen

The LPS extract was analyzed using sodium dodecyl sulfate (SDS)-polyacrylamide gel electrophoresis (PAGE) according to procedure of [23] with some modification. Briefly, about 5 μl of

LPS extract was mixed with 5 μl Laemmli sample buffer solution (2% SDS, 5% 2-mercap-toethanol, 10% glycerol, 0.0625 M Tris-hydrochloride buffer (pH 6.8), and 0.01% bromophenol blue) and heated at 100˚C for 5 min in a boiling-water bath. Then, the prepared LPS samples and 2.5% of 10 μL bovine serum albumin (BSA) as a positive control were loaded onto a 15% SDS-PAGE gel and run gel electrophoresis at 150 V for 30 min using BIORAD's Mini Protean Tetra Cell apparatus (Bio-Rad Laboratories, USA). Afterwards, the gel was stained with Coomassie blue R-250 and destined using destining solution with gentle agitation on shaker at 80 rpm for 3 hrs. The gel was visualized using visible light and photographed using digital camera for documentation and further analysis.

## Production and characterization of polyclonal antibody against EPEC LPS antigen

**i. Animals handling.** In this study, the use of mice were thoroughly approved by the College of Natural and Computational Science Institutional Review Board (CNS-IRB) under approval number of IRB/039/2019. To generate anti-LPS polyclonal antibodies and to design an immunodiagnostic assay, pathogen-free Swiss female mice (6 weeks old, 25–30 gm) were procured from Ethiopian Public Health Institute (EPHI), Addis Ababa, Ethiopia. The mice were settled in a controlled housing environment that provided them with an enriched living space, allowing unrestricted access to standard laboratory rodent food and fresh drinking water throughout their residence at animal facility of Collage of Natural and Computational Science, Addis Ababa University. To further ensure the well-being of the mice, a 14-day acclimatization period was established, allowing them to adjust to their new surroundings before experimental protocols began.

**ii. Mice sacrifice and anesthesia.** Before blood collection from the heart, all mice were humanely euthanized by the spinal cord dislocation method using CO2 anesthetic gas inhalation for a minute. This approach aimed to ensure humane treatment and to minimize pain and distress in a sterile environment, with close monitoring for any signs of discomfort. Blood collection was conducted with care to avoid complications, and the hearts of the euthanized mice were accessed through a precise incision for direct cardiac puncture. All instruments used during the procedure were sterilized in advance to preserve the quality of the collected samples. The procedure not only adhered to established ethical norms but also aimed to yield high-quality blood samples necessary for the subsequent analyses. Post-procedure, every effort was made to ensure that the care and handling of the animals were respectful, emphasizing the importance of ethical considerations in scientific research involving vertebrate species.

**iii. Immunization and blood collection.** After 14 days acclimatization, approximately 100 μl serum per mouse was collected and used as pre immune sera (negative control). The LPS antigen-adjuvant emulsion (Alum) mixture was prepared at a volume of 50 μl per mouse (1:1). Then the mice were injected with the prepared immunogen intraperitoneally. Subsequently, the 2nd and 3rd boosting immunization were carried out using the same amount of immunogen within two weeks interval. After 10 days of post immunization, the mice tails were cleaned by 70% alcohol and 100 μl of blood/mouse was collected from the tail vein each mouse using 3 ml syringe needles. Eventually, after the tertiary immunization, whole blood were collected from each sacrificed mouse [24]. The pre-immunized blood as negative control, 1st, 2nd and 3rd immune response blood sample clotted overnight at 4˚C. After that the blood sample were centrifuged at 4000 rpm at 4˚C for 10 minutes. The serum part were collected and stored at—20˚C until further use [25].

## Determination of optimal antibody titer for the development of immunodiagnosis assay

The serum was serially diluted at tenfold dilutions of $10^0$, $10^{-1}$, $10^{-2}$, $10^{-3}$, $10^{-4}$, $10^{-5}$, $10^{-6}$, $10^{-7}$, $10^{-8}$, $10^{-9}$, and $10^{-10}$, with a dilution buffer (1PBS). Diluted sera were then used for the slide agglutination assays with prepared samples of EPEC. The agglutination reactions were performed as described in the previous sections.

## Development of immunodiagnostic assay using purified LPS antigen of EPEC

EPEC LPS antigen (40 μL) and 40 μL mice sera were placed on a sterile microscopic glass slide by a micropipette and mixed thoroughly by stirring with tips. The results were read within 6–12 minutes. Slides with formed or failed agglutination were photographed with the record of the time taken for the reaction. The result was recorded as positive, if agglutination occurs, and negative, if no agglutination occurs [25].

## Development of immunodiagnostic assay for EPEC strain detection by using whole cell antigen of EPEC strain

The EPEC k 51 isolates were grown for 24 hrs on TSA at 37˚C. Then, 40 μL of serum was applied on a glass slide and 1xPBS suspended 3 colonies were added into glass slide which contain anti-LPS pAb and well mixed by using autoclaved tooth pick. The reactions were read with the naked eye by holding the slide in front of a light source against a black background (indirect illumination) within 1–12 min. A positive reaction was seen as a visible agglutination while a negative reaction was recorded if the homogeneous milky turbidity persisted [25].

## Determination of the appropriate culture medium for the newly developed assay

To determine the appropriate media for optimum agglutination of EPEC with the polyclonal serum, the EPEC k51 isolate was inoculated into EMB, TSA and TSB agar plates and incubated at 37˚C for 24 h. Then,3 colonies of EPEC k51 from EMB, TSA and 40 μl of broth culture from TSB were taken separately and tested by slide agglutination using the above procedure.

## Determination of the specificity of the newly developed immunodiagnostic assay

The pathogenic *E. coli* (10 STEC strain and 10 EPEC strain), 10 non-diarrheic and 4 non-*E. coli* (*Shigella flexneri*, *Salmonella Typhimurium*, *Staphylococcus aureus*, and *Klebsiella pneumoniae*) strains were grown for 24 hrs on TSA at 37˚C. After growth, 3 colonies, from each bacterial strain were taken and diluted in 40 μl of 1 PBS. The bacterial suspension was tested following the above procedure. For each test, the result was recorded as either negative or positive for agglutination, as very strong (+ + ++), strong (+ ++), moderate (++), (+) for fine agglutination or weak and (-) no agglutination [26].

## Evaluation of the analytical sensitivity of the newly developed immunodiagnostic assay

To evaluate the analytical sensitivity of the developed diagnostic assay in this study, the EPEC k51 strain was cultured on TSA medium and incubated overnight at 37˚C. About 40 μL of 1

×PBS serially diluted colony ($10^0$, $10^{-1}$, $10^{-2}$, $10^{-3}$, $10^{-4}$, $10^{-5}$, $10^{-6}$, $10^{-7}$, $10^{-8}$, $10^{-9}$, and $10^{-10}$) were added on the slide Then 40 μL of the polyclonal antibody against LPS of EPECK51 strain was added in to separate circles on a glass microscopic slide and carefully mixed using applicator stick and spread over the entire area enclosed by the circle. Then the slide was shacked manually by turning side to side.

### Detection performance of the newly developed immunodiagnostic assay in fecal samples

To evaluate the detection performance of the developed assay, 0.11 gm of human stool sample was spiked with 200 μl of 1×PBS suspended EPEC k51 isolate. Then ten-fold serial dilutions were performed as follows $10^0$, $10^{-1}$, $10^{-2}$, $10^{-3}$, $10^{-4}$, $10^{-5}$, $10^{-6}$, $10^{-7}$, and $10^{-8}$. Then, 40 μL of diluted spiked sample were taken from each dilution factor and tested by slide agglutination; 5 μL from the last positive agglutination result was transferred in to TSA plate to count colony forming units. The plates were incubated for 24 hours at 37°C. Then the number of colonies forming units of the selected dilution was counted and converted in to number of colony forming unit per ml which indicated the detection limit of the developed assay to detect EPEC strain from sample.

### Ethical approval

For this study, the bacterial isolates were provided by Prof. Tesfaye Sisay Tessema, AAU, and preserved in a refrigerator at –20 degrees Celsius. The human or animal bacterial samples were not isolated from this study but from the previous study. Therefore, the samples (*E. coli* isolates) used in this study were approved by the College of Natural Science Institutional Review Board (CNS-IRB) under protocol number IRB/024/2017; the project proposal for the MSc. thesis was submitted by [20]. However, we used laboratory animals, and in this case, we obtained ethical approval from the relevant authority. This study was conducted in accordance with the principles of the Declaration of Helsinki. Approval was obtained from the Institutional Review Board (IRB) Committee of the College of Natural and Computational Sciences, Addis Ababa College, Ethiopia (approval number CNS-IBR/039/2019).

### Statistical analysis

The Sensitivity, Specificity, Positive Predictive Value (PPV), and Negative Predictive Value (NPV) and accuracy were subjected to MedCalc Statistical Software version 18.11.6 (MedCalc Software bvba, Ostend, Belgium;https://www.medcalc.org; 2019) at 95% confidence interval.

## Results

### Extraction and purification of EPEC LPS antigen

The extraction process yielded 0.25 grams (250 mg) of lipopolysaccharide (LPS) from 298 ml of Enteropathogenic *Escherichia coli* (EPEC-k51) culture grown overnight in (TSB), which reached an OD600 of 0.88. The LPS was dissolved in 2 ml of Phosphate Buffered Saline (PBS), resulting in a final concentration of 125 mg/ml (Fig 1).

### Analysis of the LPS extract on SDS-PAGE and agarose gel electrophoresis

After extracting of LPS, the extracts were subjected to SDS-PAGE for determining the purity of LPS. The result depicted a single distinct band on the gel indicates that the minimal contamination on the extracted LPS. Additionally, to further verify whether the extracted LPS were

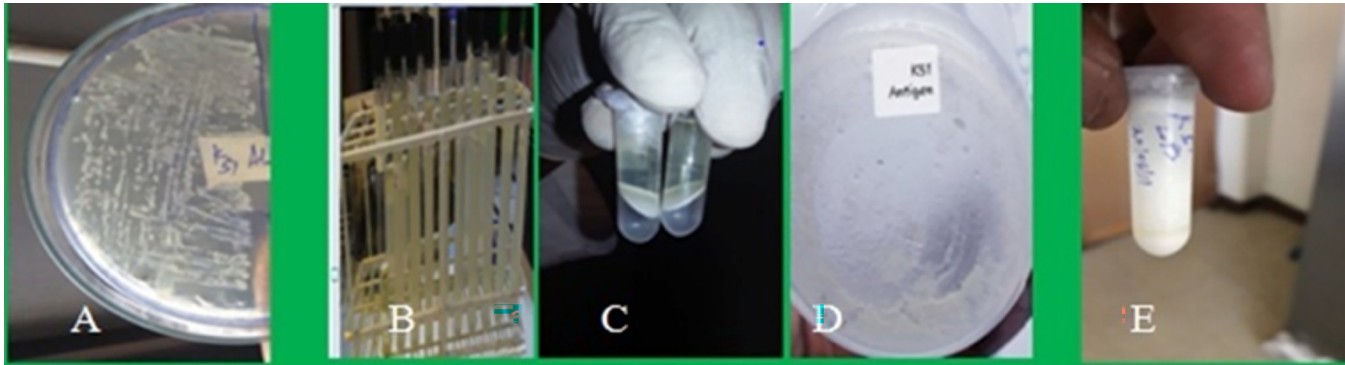

**Fig 1.** The sequential steps in the LPS preparation process are as follows: First, the EPEC strain was grown overnight on a (TSA) plate (A). Subsequently, the strain was grown overnight in TSB broth (B). After the final centrifugation, the mixture separated into three distinct layers: the chloroform layer at the top, the biomass including cell lysate in the middle, and the methanol layer at the bottom (C). The methanol and chloroform were then evaporated, leaving behind dried LPS (D). Finally, the dried LPS was weighed and dissolved in sterilized PBS to create the LPS solution (E).

contaminated with DNA and RNA contaminants, LPS extraction was subjected to agarose gel electrophoresis. The result showed that no bands on agarose gel were observed which reveals the purity of LPS (Fig 2).

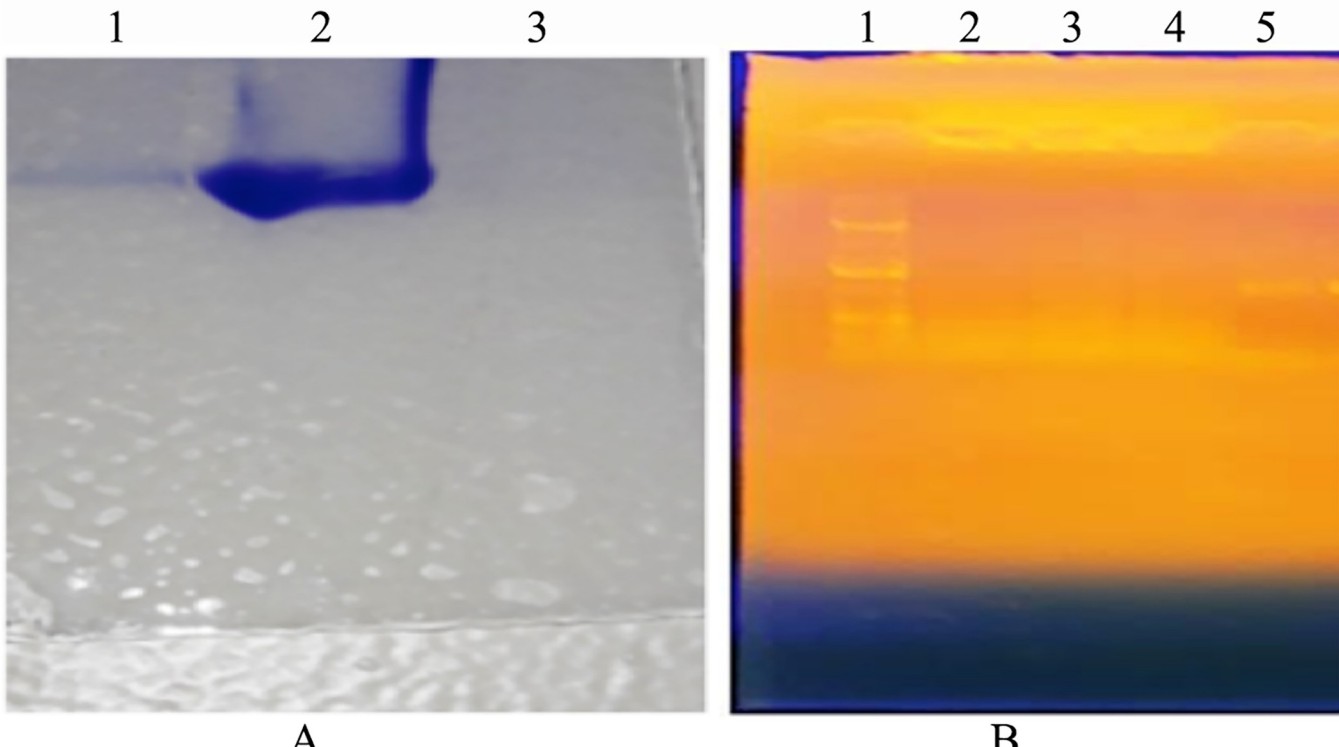

**Fig 2.** Purity analysis was conducted on the extracted LPS to determine protein contamination (A) and DNA contamination (B). For protein contamination analysis (A), a Coomassie blue-stained SDS-polyacrylamide gel was used to visualize the LPS extracted from EPEC k51. Lane 1 contained 40 μl of LPS, Lane 2 had 10 μl of BSA, and Lane 3 served as the negative control with 10 μl of buffer sample. To analyze DNA contamination (B), agarose gel electrophoresis was performed on the LPS extract. Lane 1 contained a 1 kb DNA ladder, Lanes 2–4 had 40 μl of LPS, and Lane 5 served as the positive control with 20 μl of PCR product for DNA.

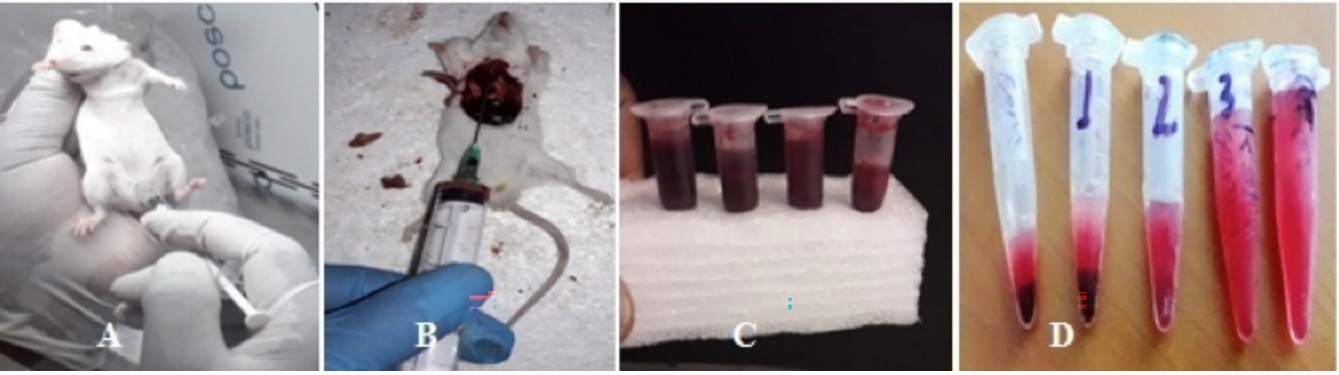

**Fig 3.** Production of Mouse Polyclonal Antibodies against EPEC Strain K-51 (A) Intraperitoneal injection of antigen preparation (LPS + alum) in the lower right quadrant using an Insulin syringe: 27 G, 1/2 in, 1.0 ml; (B) Whole blood collection of mice from cardiac puncture; (C) Collected blood before centrifugation; and (D) Separated sera at each immunization step.

## Production and characterization of polyclonal antibodies against the purified LPS antigen

Basal blood as negative control was collected before the mice were immunized with LPS. Thus, 1 ml serum was collected from 1.5 ml blood, which was drawn from tail of 40 mice. After immunization of the mice with LPS at the 1$^{st}$, 2$^{nd}$ and 3$^{rd}$ immunization on average 500 μl from tails of 23 mice, 300 μl from tails of 16 mice and 3 ml serum from heart puncture of 15 mice were obtained from each immunization step, respectively (Fig 3).

Development of the polyclonal antibody in immunized mice were identified by slide agglutination test with the immunogen (LPS). The developed antibody was revealed by agglutinates (clumps) that resulted from cross linking between molecules of LPS (antigen) and the polyclonal antibody raised against EPEC k51 LPS in the serum. The Antibody were formed in the antiserum after first booster injection and thereafter. In order to investigate the relationship between the immunization time and the immune response in a successive generation of polyclonal antibody we performed test by slide agglutination with different initial conditions based up on collected serum at time interval successively. Whereas the outcome is the difference in the amount of antibodies produced for the primary and booster immunizations. More specifically, the maximum level of antibodies relative to the injection of antigens at time interval (i.e., the boost injection), that the second immune response is faster and stronger than the first while the third is more faster and stronger than the second. The primary-immune serum, first boosting injection serum and second boosting injection serum was checked for agglutination assay respectively. However, when a pre-immune serum was used in agglutinates analysis, no corresponding clear clumps was observed which suggest that the mice were not previously exposed to these bacteria. On the other hand agglutination test using the serum that collected from primary immunization shows a slight positive reaction due to pro-inflammatory of mice to EPECk51 LPS induce innate immune response (Fig 4).

## Determination of optimal antibody titer for developing immunodiagnostic assay

The polyclonal antibodies were produced in response to immunization with EPECk51 LPS as an immunogenic. Upon repeated immunizations, the antibodies produced were sampled approximately 10 days after each boost and primary immunization. After the third injection of the antigen, both serum were checked for the production of antibodies; thus, the second and

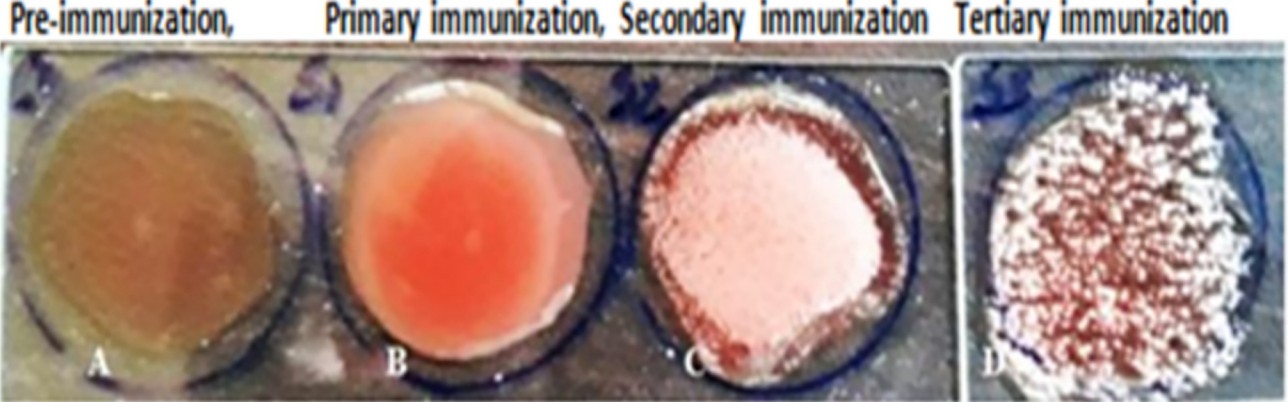

**Fig 4. Slide agglutination assay using serum collected at various times points before and after secondary, tertiary immunization and analyzed for Ab levels by slide agglutination.** A: control serum with bacteria, B: primary-immunized serum with test sample, C: second boosting serum with sample test and D: tertiary boosting serum with test sample.

third boosting showed strong and very strong agglutination. Following this, a series of tenfold dilution of the antibodies were carried out for the secondary and tertiary boosting to know variation in antibody concentrations between the two immunizations. Then the degree to which the antibody solution can be diluted and still produce detectable levels of agglutination were identified to be $10^{-3}$ and $10^{-5}$ for second and third boosting, respectively (Figs 5 and 6), showing that the level of antibody produced in serum after the tertiary immunization is much higher than that produced after the secondary immunization.

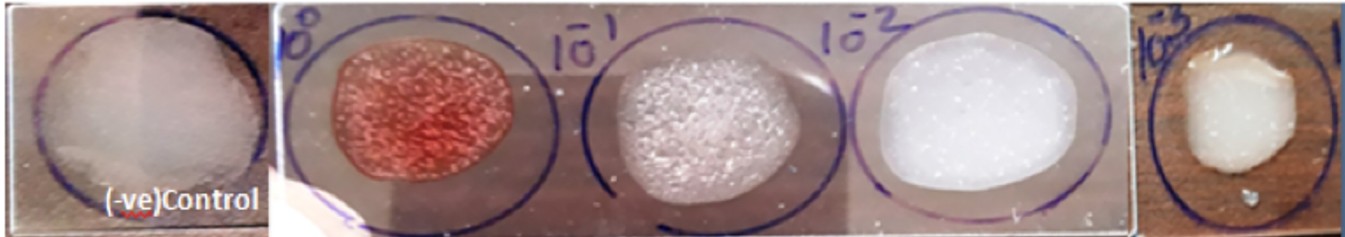

**Fig 5.** Slide agglutination assay using titers (tenfold dilution) of second boosting serum; from left to right: Negative control, $10^0$, $10^{-1}$, $10^{-2}$ and $10^{-3}$. The titer of the antibody from the tertiary immunization was found to be $10^{-3}$.

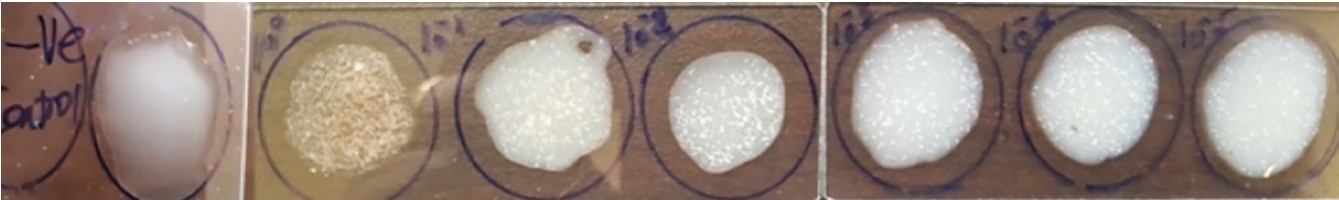

**Fig 6. Slide agglutination assay using titers (tenfold dilutions) of third boosting antibody: Negative control, $10^0$, $10^{-1}$, $10^{-2}$, $10^{-3}$, $10^{-4}$, and $10^{-5}$.** The titer of the antibody from the tertiary immunization was found to be $10^{-5}$.

**Table 1. Optimization of slide agglutination assay for detection of anti- EPEC isolate k51 serum.**

| Tested components | Serum before immunization | Serum from 1st immunization | Serum from 2nd immunization | Serum from 3rd immunization | Detection time (min) |
|---|---|---|---|---|---|
| EPEC of LPS extract (antigen) | - | + | ++ | +++ | 5–15 |
| Undiluted colony of EPECK51 | - | - | ++ | +++ | 4–12 |
| Colony diluted with 1×PBS | - | + | +++ | ++++ | 2–6 |

*'+' Positive result;'–' Negative result. EPEC-Enteropathogenic Escherichia coli, LPS- Lipopolysaccharide, PBS-Phosphate Buffer saline.

## Development of immunodiagnostic assay using purified LPS and whole cell antigen of EPEC

In this study the slide agglutination immunodiagnostic assay was developed using the purified LPS, whole cell colony, and PBS suspended whole cell colony of EPEC strain as antigen and anti-LPS pAb. Thus, the EPEC (k51) LPS, colony of EPEC k51 suspended in 1% PBS and direct loop full colony of EPEC k51 were tested. A positive agglutination reaction occurred within average 5–15 minutes for LPS and the agglutination for colony suspension in 1% PBS and direct loop full colony was fully developed within 2 to12 minutes in all types of tested antigens in a different serum. The strength of positive agglutination was assigned as weak (1+), moderate (2+), strong (3+) or very strong (4+) and no agglutination (-) (Table 1).

## Development of immunodiagnostic assay using purified LPS antigen of EPEC for whole cell

To evaluate the functionality of the developed assay, direct colony and colony suspension in 1×PBS analysis was conducted using sera that were collected after second and third boosting. We observed visible agglutination in both direct colony and colony suspension, suggesting that the developed assay works well. However, the degree and rapidity of agglutination was not consistent in each test. Compared to direct colony, diluted colony method was accurate and clear due easily dispersed in solution enhance the reaction in detecting EPEC infection (Fig 7). Taking all together, the agglutination tests using bacteria colony and LPS with the collected serum indicates the produced antibodies are specifically react to the LPS antigen of EPEC K51 isolate.

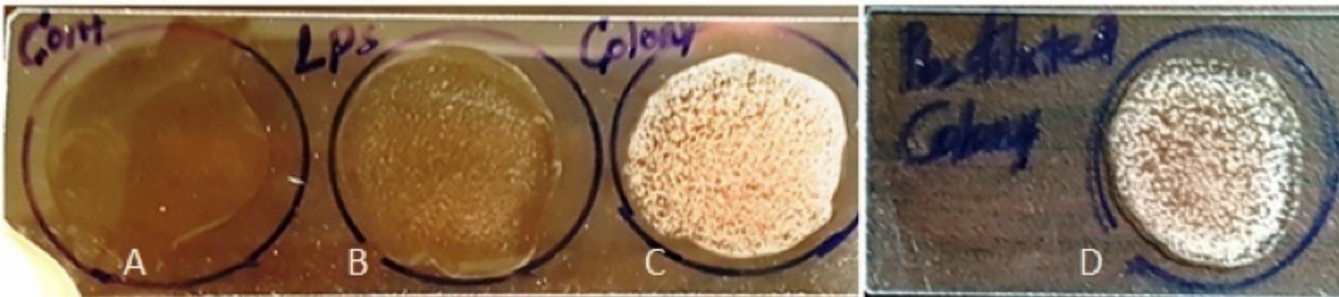

**Fig 7.** Optimization of agglutination assay by using tertiary boosting serum with equal amount of test sample (EPEC K51) A: control serum with LPS, B: LPS with serum, C: Un suspended colony with serum D: colony suspension with serum.

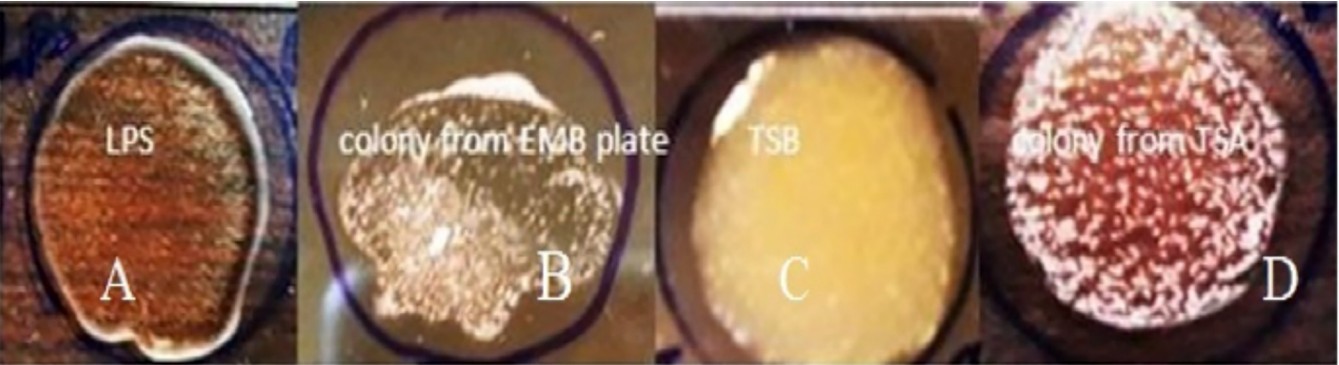

**Fig 8. Agglutination of EPEC grown on different culture media and agglutination performance with antibody raised against LPS of EPEC.** A. extracted LPS with sera showed faint agglutination (+), B.colony from EMB medium agglutination (+++), C. colony from TSB low agglutination(++), D. colony from TSA high agglutination (++++).

## Comparison of appropriate bacterial culture medium for performing agglutination assay

Furthermore, in order to assess the effect of culture media on the performance of the newly developed agglutination assay through affecting the antigen-antibody reaction, bacteria to be used for the assay were grown in three culture media commonly used for *E. coli* growth, EMB agar, TSA and TSB; the agglutination performance showed that overnight colony from EMB, TSA and broth culture from TSB produced medium (++), high (++++) and low (+) levels of agglutination, respectively (Fig 8).

## Determination of the specificity of the newly developed immunodiagnostic assay

The specificity of the developed slide agglutination assay was evaluated by using a total of 34 bacteria (20 pathogenic *E. coli* isolates, 10 non diarrheagenic isolates and 4 non *E. coli* standard strains (*Klebsilla pneumonia*, *Shigella flexinari*, *Salmonella Typhimurium* and *Staphylococcus aureus*)). Fixed amount of serum that contains antibody was added to equal amount of PBS dissolved colony for all specificity test of this assay. The slide agglutination assay was highly specific, agglutination formed with EPEC isolates and was not cross-reactive with other bacterial strains that included in this study except one isolate (D 24) from STEC strains. However, the degree (positivity) and the agglutination time of EPEC strain was valid from this false positive strain and (Figs 9, 10) and (Table 2).

Moreover, according to MedCalc Statistical Software diagnostic assay evaluation test result, the assay showed 100% sensitivity; 10 of 10 EPEC strains showed positive agglutination with the pAb against EPEC strain, 95.83% specificity; 23 of 24 non EPEC strains (10 STEC, 10 *E.coli* isolated from Non diarrheic calves and 4 non *E. coli* spp) showed negative agglutination with the pAb serum against EPEC strain (Table 3).

In addition, the assay deciphered with 99.79% accuracy, 99.78% negative predictive value and 100% positive predictive value. The 100.0% positive predictive value (PPV) indicate the probability that 10 EPEC isolates were tested positive is in fact positive with regard to the true diagnostic kit while 95.833% negative predictive value (NPV) is the probability that 10 STEC strain, 10 non diarrheic isolates and 4 non *E. coli* (*Shigella flexineri*, *Salmonella Typhimurium*, *Staphylococcus aureus*, and *Klebsiella pneumoniae*) isolates were tasted negative is in fact

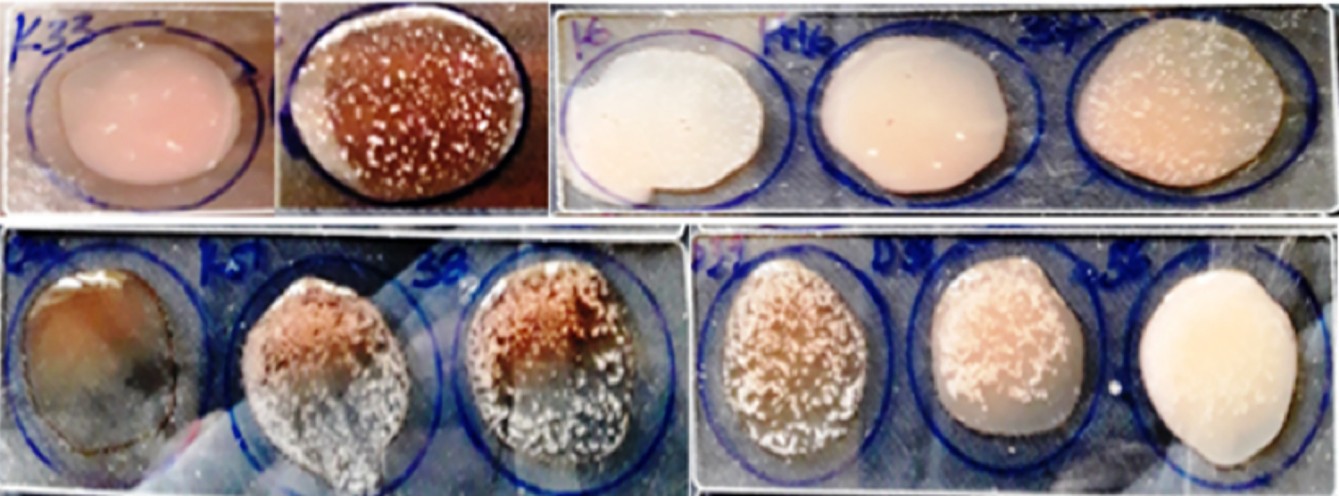

**Fig 9. Results of agglutination for 10 EPEC strain isolates.**

negative with regard to the true diagnostic status and 97% of accuracy was an indication of the extent to which a test confirms to truth (Table 4).

## Determine the sensitivity of the newly developed slide agglutination immunodiagnostic assay

To measure sensitivity of the developed assay, EPEC strain was cultured on TSA medium overnight at 37°C and single colony was picked and incubated in TSB for 24 hours at 37°C. 100 to 10–10 serial dilutions as tenfold serial dilutions of EPEC strain were carried out through adding 1 μL of diluted colony of EPEC strain and broth culture in 180 μL of 1% PBS containing test tube. This step was repeated for consecutive test tubes with appropriate serial dilution mix. Subsequently, 40 μL of isolate suspension was taken from each dilution factor and added to microscopic glass slide. 40 μL of serum polyclonal antibody against LPS of EPEC strain was

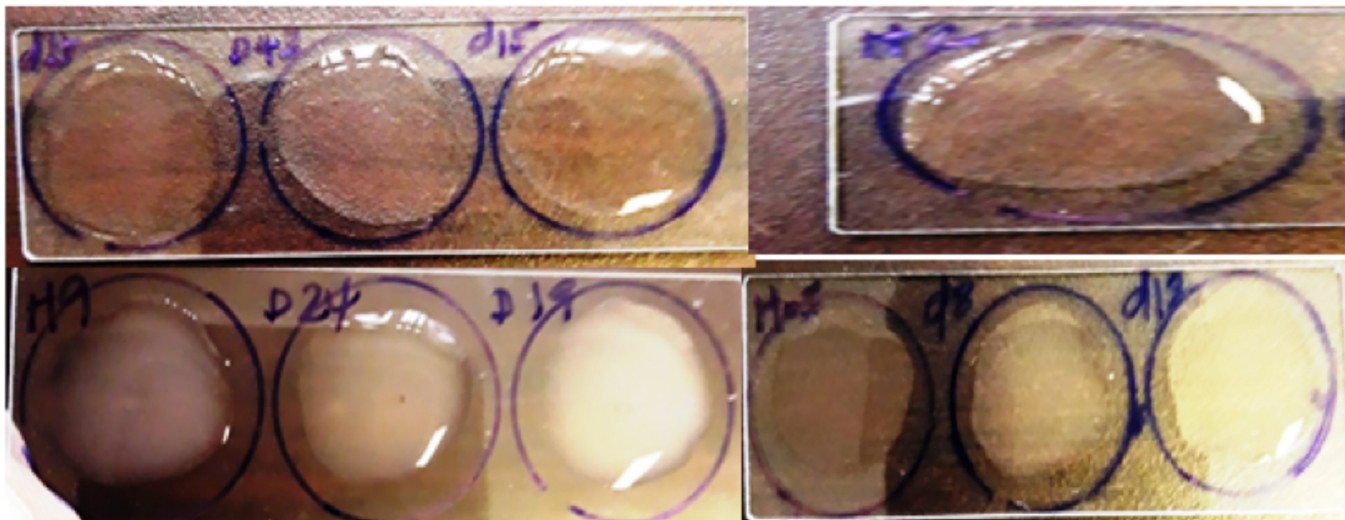

**Fig 10. Results of agglutination for 10 STEC strain isolates.**

**Table 2. The result of the specificity test of the assay using different strains.**

| Bacterial Categories | code | EPEC (+ve) Serum | Min -Max detection in minutes |
|---|---|---|---|
| STEC | H2 | - | 5–12 |
| | H9 | - | 5–12 |
| | D27 | - | 5–12 |
| | D29 | - | 5–12 |
| | D19 | - | 5–12 |
| | D24 | + | 5–12 |
| | D8 | - | 5–12 |
| | D43 | - | 5–12 |
| | D23 | - | 5–12 |
| | H05 | - | 5–12 |
| EPEC<br>*E. coli* isolated from Non-diarrheic calves | D33 | ++++ | 2–6 |
| | D39 | +++ | 2–6 |
| | 56 | ++ | 5–10 |
| | 38 | ++++ | 2–6 |
| | 34 | ++ | 5–10 |
| | K33 | + | 5–12 |
| | K51 | ++++ | 2–6 |
| | 16 | ++ | 5–10 |
| | H16 | + | 5–12 |
| | D44 | ++++ | 2–6 |
| | N107 | - | 5–12 |
| | N108 | - | 5–12 |
| | N154 | - | 5–12 |
| | N156 | - | 5–12 |
| | N157 | - | 5–12 |
| | N158 | - | 5–12 |
| | N159 | - | 5–12 |
| | N160 | - | 5–12 |
| | N155 | - | 5–12 |
| | N166 | - | 5–12 |
| Non E. coli | Shigella | - | 5–12 |
| | Salmonella | - | 5–12 |
| | *Staphylococcus aureus* | - | 5–12 |
| | *Klebsiella pneumoniae* | - | 5–12 |

*The strength of agglutination was assigned as weak (1+), moderate (2+), strong (3+) or very strong (4+) and no agglutination (-). EHEC -Enterohemorrhagic Escherichia coli, EPEC-Enteropathogenic Escherichia coli, STEC -Shiga toxin Escherichia coli.

added into each glass slide that contained suspension of EPEC bacteria. After solutions were mixed carefully, the slides were tilted gently by rotating with hand. From $10^0$ to $10^{-6}$erial dilutions were detected by this assay. However, the appearance of agglutination was taken slightly

**Table 3. Diagnostic sensitivity and specificity estimates calculated from samples of known 10 EPEC positive samples and 24 known negative samples.**

| Test | Present | N | Absent | N | Total |
|---|---|---|---|---|---|
| Positive | True Positive | a = **10** | False Positive | c = **1** | a + c = **11** |
| Negative | False Negative | b = **0** | True Negative | d = **23** | b + d = **23** |
| Total | | a + b = **10** | | c + d = **24** | **34** |

**Table 4. MedCalc statistical software version 18.11.6 results of sensitivity and specificity.**

| Statistic | Value | 95% CI |
|---|---|---|
| Sensitivity | 100.00% | 69.15% to 100.00% |
| Specificity | 95.83% | 78.88% to 99.89% |
| Positive Likelihood Ratio | 24.00 | 3.52 to 163.50 |
| Negative Likelihood Ratio | 0.00 | |
| Disease prevalence (*) | 95.00% | |
| Positive Predictive Value (*) | 99.78% | 98.53% to 99.97% |
| Negative Predictive Value (*) | 100.00% | 85.18% to 100.00% |
| Accuracy (*) | 99.79% | |

longer time (up to 12 minutes) when the dilution factor was increased. In the light of this result we want extend the evaluation of the assay to determine the number of CFU ml$^{-1}$ of the selected dilution. To this end, 5 μL aliquot of $10^{-6}$ dilution factor was transferred into TSA plate for counting of colony forming units; thus, 68/5 μL colonies were counted; this corresponds to13.6 X10$^{6}$CFU ml$^{-1}$. The results reveals that the devised assay can able to detect up to 13.6 X10$^6$ CFU ml$^{-1}$numbers of EPEC (as a minimal CFU/ml) were present in a given sample (Fig 11).

## Detection performance of the newly developed immunodiagnostic assay in fecal samples

To determine the applicability of the developed slide immunodiagnostic assay for detection of EPEC strain in fecal sample, human feces were collected and spiked with EPEC in TSB broth

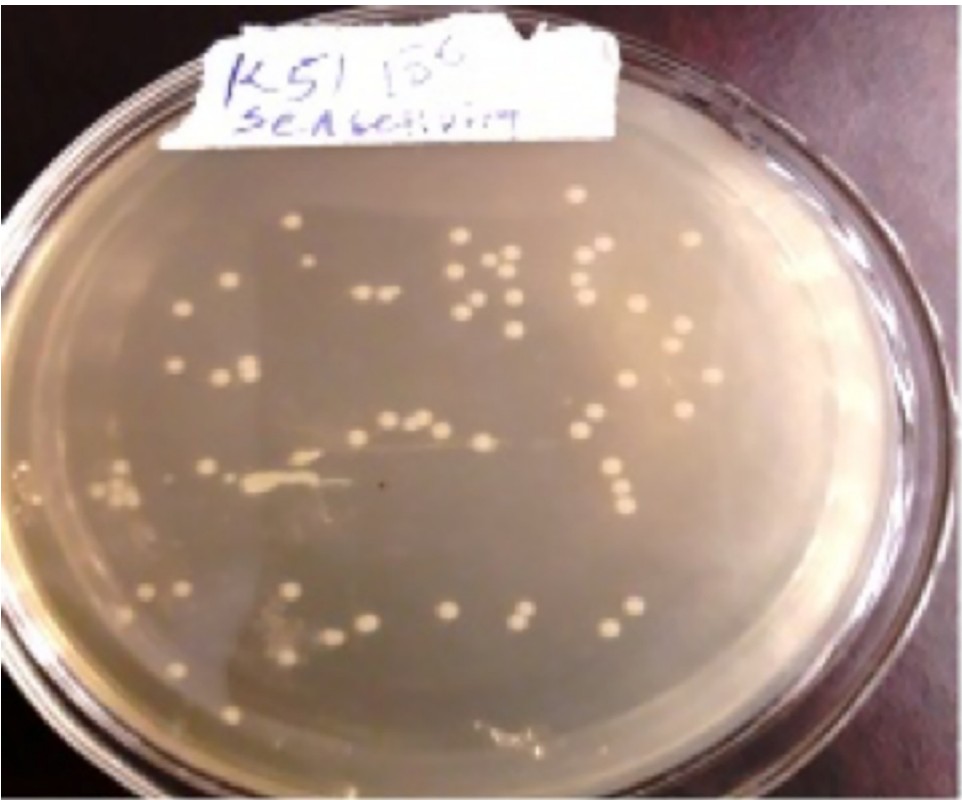

**Fig 11. Colony forming units of EPEC K51 strain ($10^{-6}$ dilution) grown on TSA plate.**

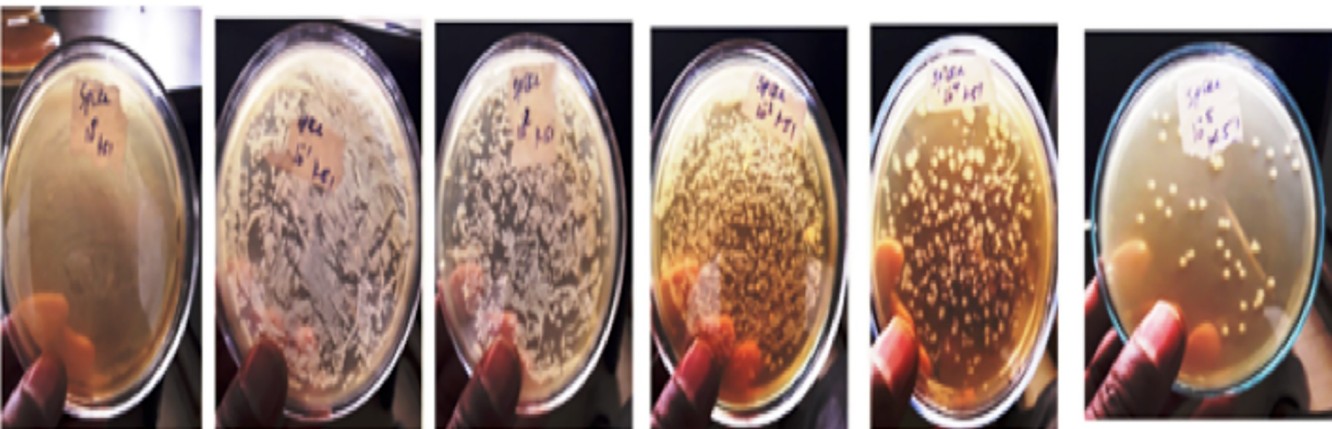

**Fig 12. Colony forming units grown on TSA from serial tenfold dilution ($10^0$ to $10^{-5}$ dilutions) of fecal sample spiked with EPEC K51 strain.**

culture as mentioned in material and method. First, the spiked sample was tested on slide agglutination and the result showed that the slide agglutination assay developed in this study detected spiked EPEC. To compare detection methods, culture-based assays were also performed. For these assays, spiked fecal samples were plated, and EPEC colony growth was monitored. Some EPEC strains showed initial growth after approximately 8 hours of incubation, while the majority achieved visible growth within 24 hours, demonstrating the capability of traditional culture methods to support EPEC growth under standard conditions. Secondly, to determine the detection limit of the developed assay for the spiked EPEC strain, serially diluted spiked EPEC ($10^0$ to $10^{-9}$ dilution factors) was used for slide agglutination test. In this study the result showed that the assay detected up to $10^{-5}$ dilution. From this minimal dilution, 5 µL of aliquote was transferred to TSA medium for CFU counting; hence 52 colonies /5 µL were counted, indicating that 104X $10^4$ CFU ml $^{-1}$ EPEC from fecal sample could be detected (Fig 12).

## Discussion

Enteropathogenic *E. coli* (EPEC) is one of the very paramount human pathogens and is responsible for morbidity and mortality in infants especially in the developing world and in immunocompromised individuals. Therefore, to overcome this global health problem, WHO recommended Accurate Test-and-Treat Strategy (ATTS). Nowadays, several diagnostic assays for EPEC have been developed based on different type antigens as biomarker [8]. However, the high diversity of these antigens hinders to apply ubiquitous EPEC diagnostic assay worldwide. Moreover, the antigenic variation of EPEC strains throughout world lead to low diagnostic value in both sensitivity and specificity and no longer consider as efficient method, nor a reliable one especially, when there is discrepancy between the origins of the strain and the serum sample to be tested. Molecular techniques such as PCR assays provide an efficient and convenient method for strain serotyping in microbiology labs, particularly in regions where PCR is routinely used for identifying EPEC strains. However, detecting genes does not guarantee virulence factor expression, and the genetic diversity of strains complicates molecular detection, making it reliant on primer quality and technical expertise [27, 28]. In many areas with limited molecular technology, diagnostic labs still depend on traditional culturing techniques for EPEC identification [29]. This method is time-consuming (8–24 hours) and may delay appropriate treatment, potentially exposing patients to antibiotics, which could

contribute to the development of multidrug-resistant (MDR) infection [30]. This study developed immunodiagnostic assay by targeting LPS immunodominant antigen to facilitate the rapid, cost-effective, sensitive, and specific diagnosis of EPEC infection. A slide agglutination immunodiagnostic assay was developed for the detection of enteropathogenic EPEC K51 using the purified LPS antigen and anti-LPS polyclonal antibody as a paramount component for the detection of EPEC seropositive and EPEC stool samples. This assay offers a practical alternative for EPEC detection, aiming to improve diagnostic efficiency and accuracy in clinical and laboratory settings.

To determine and identify the optimal sample type, specific EPEC antigen (LPS), EPEC whole cell antigen (overnight culture colony) and diluted EPEC whole cell antigen (PBS suspended overnight culture colony) were evaluated by the newly developed EPEC immunodiagnostic assay. The results revealed that the slide agglutination reaction rate between the purified single LPS antigen and the purified anti-LPS polyclonal mice antibody was moderate (Fig 7B), while the agglutination rate of mice anti-LPS pAb with overnight culture whole cell colony EPEC antigen was strong (Fig 7C), but the agglutination rate were affected with the antigen status such as the bacterial growth phase, growth characteristics and colony size, colony morphology and number of colony used.

In the case of slide agglutination between the anti-LPS PAb with PBS suspended overnight culture EPEC whole cell antigen revealed very rapid and strong agglutination reaction (Fig 7C). This is because the physiological saline solution aids to reduce inhibiters and non-specific binding that enhance the antigen-antibody reaction, but the intensity of the agglutination reaction may vary according to the density of the cell suspension or the amount of antiserum used. As expected, this result agreed with the previously reported slide agglutination immunodiagnostic assay for STEC strain detection.

The variations in strength of agglutination between specific LPS extract and whole cell colonies can be explained immunologically as follows: LPS is a subunit structural component of the bacteria, so when the antibodies cross link many LPS molecules agglutination will be formed but the strength is limited by the small molecular size of the antigen used (LPS). On the other hand, when using the whole bacterial cell as an antigen, many LPS antigens on the cell surface (a cell of about 0.6 μm for *E. coli*) make a cross link binding with the purified specific polyclonal antibodies, which lead a much better visible agglutination results. Furthermore, the expression level of the EPEC bacterial strain cell surface LPS antigen was evaluated on TSA, EMB plates and TSB broth growth medium condition and the result showed that the agglutination rate between the EPEC whole cell antigen obtained from TSA growth medium was very strong and effective than TSB and EMB plates, which indicated that the TSA growth medium is an ideal medium for the highly expression of LPS antigen of EPEC strains [31, 32]. When considering the effects of EMB media on the growth of EPEC, it is worth because colony was too small in size and reaction with generated PAb reduced in agglutination rate.

Antibody-based immunoassays are rapid, but most assays performed to detect bacterial species are performed after the test samples are enriched for 24 h on media [33]. However, these media may or may not support the expression of the antigen of interest that was used as target for immunoassays, since each component of the media can interfere with the regulation of gene expression in bacterial pathogens. According to the current study results, overnight cultures that had stored more than 24 h at 4°C sometimes produced various results; therefore, the agglutination tests should be performed on fresh cultured organisms, which in lined with the previous finding [34].

It was interesting that degree and speed of agglutination was found to vary among antibodies collected from the primary, secondary and tertiary immunizations, suggesting that the affinity of antibodies generated and population of LPS specific memory B cells were increase

during subsequent immunizations. This due to the result of the memory B cells responded rapidly and expanded into plasma cell populations in subsequent exposures of the mice to EPEC k51 LPS. A study showed that the increased antibody levels observed in secondary and tertiary immunizations corresponds to the increased populations of plasma cells in the mice

The polyclonal antibody that was collected from first immunization was limited response for LPS and was taken a bit while, 12 min to observe the maximum clamp. Little differences were identified between agglutination times due to low concentration of polyclonal antibody in pre-serum results in weaker agglutination; such variations were also observed among bacterial isolates belonging to the same EPEC strain (Fig 10). The variation in the degree and rapidity of the slide agglutination assay among isolates of the same strain reveals differences in the amount [35].

Since the concentration of antibody in a serum cannot be known by slide agglutination, the determination of antibody titer among the antibodies from the three immunizations was determined using serial tenfold dilution. Titers of $10^{-3}$ and $10^{-5}$ were detected for antibodies from secondary and tertiary immunization, respectively, suggesting that the produced polyclonal serum was highly concentrated and had strong affinity. Due to the collected serum from each mouse was limited to carry out all performance tests of the assay, we made pooled sera from group of animals immunized with the same antigen and eventually pooled sera was dissolved with 1×PBS.

The specificity of the newly developed slide agglutination immunodiagnostic assay using antiserum raised against LPS of EPEC k51 strain was evaluated by testing different bacteria including EPEC, STEC, non-diarrheic *E. coli* and non *E. coli* standard strains. The result of this study showed that all EPEC that were used in the study detected positive and the remaining bacteria were negative except one STEC isolate (D24), which showed weak positive agglutination indicating false positive result. Another finding also showed that the cross reactivity between EPEC and STEC serotypes in latex agglutination method [32], which suggesting that the two serotypes, EPEC and STEC share many antigen together and due to the presence of minor cell membrane protein [22, 36]. This cross reactivity might be happened due to biological diversity of EPEC pathogen, and nonspecific binding to the fc region of IgG highlighting the potential for shared antigenic properties between these serotypes.

The 2x2 tables statistical analysis performed using MedCalc Statistical Software version 18.11.6 provides a comprehensive assessment of the immunodiagnostic assay's performance in terms of sensitivity, specificity, and related diagnostic metrics. In this analysis, subjects are categorized according to a gold standard (PCR) or reference method in columns and according to test results in rows, as shown in Table 4. This format allows for a straightforward calculation of key performance measures, providing a clear picture of the assay's accuracy, predictive values, and likelihood ratios in identifying true positives and true negatives, offering insights into its diagnostic effectiveness

Taken all together, results of this study showed that 10 (100%) of 10 EPEC strains were positive for the polyclonal antibody serum against LPS of EPECK51 strain, while 9 (90%) of 10 STEC isolates, 10 (100%) of 10 non diarrheic isolates and 4 (100%) of 4 non *E. coli* strains were negative for the polyclonal antibody serum against LPS of EPEC strain. Moreover, according to diagnostic performance analysis test, the devised assay provided promising results to apply it as diagnostic method for detection of EPEC. This assay depicted specificity 95.83%, a sensitivity 100.0%, and 97% accuracy for detection of EPEC isolates. Therefore, our results clearly demonstrated that the polyclonal antibody against the EPEC k51 LPS as antigen shows high sensitivity and specificity and was be able to detect the antigen at low dilution factor by using slide agglutination analysis. The sensitivity analysis of the designed assay showed that EPEC strain could be detected by the newly developed slide agglutination assay at a level of $10^{-1}$–

$10^{-6}$ CFU∕ml of EPEC when pure cultures were analyzed. Thus, positive samples can be identified correctly if the positive bacterial cell material accounts for $10^{-6}$ of the total cell material.

The detection performance of the newly developed immunodiagnostic assay was assessed in detection of EPEC within fecal sample, EPEC negative fecal sample was spiked with EPEC strain. Accordingly, the spiked sample could be used for slide agglutinations. The detection limit of the spiked stool was at level of $10^{1}-10^{5}$ from serially diluted sample of stool spiked with EPEC. When cultured on TSA 52 colonies/5 μL of diluted sample were grown, indicating the presence of 104 $X10^{4}$CFU/ ml of stool sample. These results imply that the antibody is suitable for the detection of the EPEC by using slide agglutination method within the infected fecal sample with few load of EPEC strain. This is very important finding that show the applicability of the developed assay in different area like research center, hospitals and clinical setting. Clinically, it provides a rapid, sensitive, and specific means of pathogenic EPEC detection, which can support timely diagnosis and treatment decisions. In public health, the assay's quick and reliable results make it suitable for large-scale disease surveillance and outbreak management, especially in resource-limited settings. Its adaptability to detect various pathogens enhances its value in monitoring infectious diseases, contributing to effective disease control and prevention efforts. However, further studies to improve and validate the newly developed assay using a larger positive and negative sample size including comprehensive assessment and comparison of assay sensitivity, specificity and reproducibility with the commercially available gold standard EPEC strain detection kit will be required.

## Conclusion

This study has successfully developed a rapid method for detecting the most significant EPEC strains using a slide agglutination assay. The method was tested on artificially spiked fecal samples, and the results demonstrated its superior sensitivity compared to conventional microbiological methods. The study also identified the most effective media for culturing EPEC for LPS extraction and detection, with the best outcomes achieved by suspending three colonies in 1×PBS. Furthermore, the assay exhibited excellent specificity and sensitivity. Therefore, our study suggests that the designed assay, which employs anti-LPS antisera, is highly suitable for the initial screening of the majority of diarrheic EPEC infections. Notably, this assay is relatively simple, inexpensive, and can be easily produced using conventional animal-based antibody production strategies. It can be implemented in hospitals, local clinics, and research centers for the rapid detection of EPEC.

## Acknowledgments

We would like to thank W/o Amelework Eyado for her technical assistance during some of the experiments.

## Author Contributions

**Conceptualization:** Tesfaye Sisay Tessema.

**Data curation:** Aliyi Hassen Jarso, Biniam Moges Eskeziyaw, Degisew Yinur Mengistu.

**Formal analysis:** Aliyi Hassen Jarso, Biniam Moges Eskeziyaw, Degisew Yinur Mengistu, Tesfaye Sisay Tessema.

**Funding acquisition:** Tesfaye Sisay Tessema.

**Investigation:** Aliyi Hassen Jarso, Tesfaye Sisay Tessema.

**Methodology:** Aliyi Hassen Jarso, Biniam Moges Eskeziyaw, Degisew Yinur Mengistu.

**Project administration:** Tesfaye Sisay Tessema.

**Resources:** Aliyi Hassen Jarso, Tesfaye Sisay Tessema.

**Software:** Aliyi Hassen Jarso.

**Supervision:** Tesfaye Sisay Tessema.

**Validation:** Aliyi Hassen Jarso, Biniam Moges Eskeziyaw, Degisew Yinur Mengistu, Tesfaye Sisay Tessema.

**Visualization:** Aliyi Hassen Jarso, Biniam Moges Eskeziyaw, Degisew Yinur Mengistu, Tesfaye Sisay Tessema.

**Writing – original draft:** Aliyi Hassen Jarso, Biniam Moges Eskeziyaw, Degisew Yinur Mengistu, Tesfaye Sisay Tessema.

**Writing – review & editing:** Aliyi Hassen Jarso, Biniam Moges Eskeziyaw, Degisew Yinur Mengistu, Tesfaye Sisay Tessema.

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
