## [Decision Letter · Decision Letter 0]

1 Oct 2024

PONE-D-24-30559Designing of immunodiagnostic assay using polyclonal antibodies for detection of Enteropathogenic Escherichia coli strainsPLOS ONE

Dear Dr. Yinur Mengistu,

Thank you for submitting your manuscript to PLOS ONE. After careful consideration, we feel that it has merit but does not fully meet PLOS ONE’s publication criteria as it currently stands. Therefore, we invite you to submit a revised version of the manuscript that addresses the points raised during the review process.

We look forward to receiving your revised manuscript.

Kind regards,

Furqan Kabir

Academic Editor

PLOS ONE

Journal Requirements:

4. Thank you for stating the following financial disclosure: “This study was supported by the Bio and Emerging Technology Institute (BETin), Prof Tesfaye Sisay Tessema won research grant.”

5. Please provide a complete Data Availability Statement in the submission form, ensuring you include all necessary access information or a reason for why you are unable to make your data freely accessible. If your research concerns only data provided within your submission, please write "All data are in the manuscript and/or supporting information files" as your Data Availability Statement.

7. Please include a copy of Table 4 which you refer to in your text on page 16.

Reviewers' comments:

Reviewer's Responses to Questions

**Comments to the Author**

1. Is the manuscript technically sound, and do the data support the conclusions?

Reviewer #1: Partly

Reviewer #2: Yes

Reviewer #3: Yes

2. Has the statistical analysis been performed appropriately and rigorously? 

Reviewer #1: I Don't Know

Reviewer #2: I Don't Know

Reviewer #3: Yes

3. Have the authors made all data underlying the findings in their manuscript fully available?

Reviewer #1: No

Reviewer #2: Yes

Reviewer #3: Yes

4. Is the manuscript presented in an intelligible fashion and written in standard English?

Reviewer #1: No

Reviewer #2: Yes

Reviewer #3: Yes

5. Review Comments to the Author

Reviewer #1: Immune diagnosis using polyclonal sera doesnot give any novelty in diagnosis of bacterial diseases. Authors used crude antigen for raising hyperimmune sera, which can always invite nonspecific/false diagnosis of bacterial diseases.

Suggestion

1. Better to go for recombinant protein based diagnosis

2. It is necessary to validate the assay with reference methods

3. Need to illustrate relative accuracy, specificity, sensitivity of the assay

Reviewer #2: The manuscript demonstrates a well-organized structure and a lucid presentation of the authors' research on the development of an immunodiagnostic assay for detecting EPEC strains. It provides a detailed description of the methods used and evaluates the assay's specificity and sensitivity across different bacterial strains.

• The manuscript has some areas that could be improved.

• It would be beneficial to have a more detailed discussion of the study's limitations, such as the potential drawbacks of using a polyclonal antibody and the likelihood of cross-reactivity with other bacterial strains. Additionally, a more comprehensive description of the statistical analysis used to assess the assay's specificity and sensitivity would enhance the manuscript.

• The figures and tables could also be improved for better clarity and readability. Furthermore, a more detailed discussion of the potential applications of the assay in clinical and public health settings would be valuable.

Reviewer #3: Comment to authors

I am pleased to have had the opportunity to review this manuscript due to the importance of the study, which aims to develop a diagnostic technique for E. coli, a neglected but significant bacterial pathogen. This technique could greatly improve the speed of diagnosis and treatment of E. coli infections, making it a valuable contribution to patient care.

General comments

#1. Given the availability of multiple advanced options in today’s world, including molecular techniques, it would be valuable for the authors to discuss the potential gaps in their study and how it compares to these more advanced diagnostic methods. A discussion of this could strengthen the manuscript by providing a clearer understanding of the study’s relevance and limitations in the context of current diagnostic advancements.

#2. The manuscript contains several typographical and formatting errors that affect its readability. I recommend addressing the spacing issues, numerical format corrections, and inconsistencies in citation and scientific name formatting, as well as correcting other similar typographical errors.

#2.1. Specific comments;

Lines 23, 36, 64, 78, 218, 430, 499, and 507: E. coli should be italics “E. coli”

Lines 32; 36, 233, 317, 379, and 380: Escherichia coli should be italics to “Escherichia coli”

Line 37: infants (Kaur and Dudeja 2023) needs spacing “infants (Kaur and Dudeja 2023)”

Line 41: centers (Dupont et al. 2016; Gismero-Ordoñez et al. 2002) needs spacing “centers (Dupont et al. 2016; Gismero-Ordoñez et al. 2002)”

Line 77: “((Gomes et al. 2016)” delete one of the brackets “(Gomes et al. 2016)”

Lines 107, 126, and 163: -20°C spacing “- 20 °C”

Line 107: laboratory(Zenebe et al. 2024 ; Wolde et al. 2022) spacing “laboratory (Zenebe et al. 2024 ; Wolde et al. 2022)”.

Line 109: Spacing needed “37 °C”

Line 109: 300ml spacing required “300 ml”

Lines 116 and 162: 4°C spacing “4 °C”

Lines 117 and118: 2ml spacing “2 ml”

Line 121: 333µl:666µl spacing “333 µl : 666 µl”

Line 124: 50ml needs spacing “50 ml”

Line 131: 5µl needs spacing “5 µl”

Line 133: 100°C needs spacing “100 °C”

Line 138: 80rpm needs spacing “80 rpm”

Line 142: 25-30gm needs spacing “25-30 gm”

Line 151: 50μl needs spacing “50 μl”

Line 155: 3ml needs spacing “3 ml”

Line 157: 100μl needs spacing “100 μl”

Line 160: to(Joyce et al. 2014) needs spacing “to (Joyce et al. 2014)”

Line 162: 4oc needs correction “4 °C”

Line 165, 201, and 208: 100, 10-1, 10-2,10-3 10-4,10-5 ,10-6, 10-7 , 10-8,10-9,10-10 should be corrected as “100, 10-1, 10-2, 10-3, 10-4, 10-5, 10-6, 10-7, 10-8, 10-9, and 10-10”

Line 173: occurs(Moges et al. 2022) needs spacing “occurs (Moges et al. 174 2022)”

Line 180: 1-12min needs spacing “1-12 min”

Line 182: persisted(Moges et al. 2022) needs spacing “persisted (Moges et al. 2022)”

Line 197: Then,3 colononies needs spacing “Then, 3 colonies”

Line 195: 40µl need spacing “40 µl”

Line 197: agglutination(Danielsson and Kronvall 1974) needs spacing “agglutination (Danielsson and Kronvall 1974)”

Line 210: at37 ºC needs spacing “at 37 °C”

Line 229: 95%confidence needs spacing “95% confidence”

Line 260: 500µl needs spacing “500 µl”

Line 260: 23 mice,300 µl needs spacing “23 mice, 300 µl”

Line 261: 3ml needs spacing “3 ml”

Line 261: (15 mice) delete brackets “15 mice”

Line 262: (fig) which figure? 1, 2, or 3? I think the authors are saying figure 3.

Line 306: 100 ,10-1 ,10-2 ,10-3 ,10-4 ,and 10-5 should be corrected as “100 , 10-1 , 10-2, 10-3, 10-4, and 10-5”

Line 320: immunodiagnostic assay should be bold “immunodiagnostic assay”

Line 413: EPEC(as a minimal CFU/ml ) needs spacing “EPEC (as a minimal CFU/ml )”

Line 424: colonies/5 μL needs spacing “colonies / 5 μL”

Line 434: biomarker(Mare et al. 2021a) needs spacing “biomarker (Mare et al. 2021a)”

Line 444: antigen(overnight culture colony) needs spacing “antigen (overnight culture colony)”

Line 467: strains(Hahm and Bhunia 2006 ; Rocha et al. 2014).. needs spacing and delete one the full stops “strains (Hahm and Bhunia 2006 ; Rocha et al. 2014).”

Line 476: finding(Medina et al. 2012) needs spacing “finding (Medina et al. 2012)”

Line 489: antibody(Hampson 1991) needs spacing “antibody (Hampson 1991)”

Line 502: method( Rocha et al. 2014b) needs spacing “method ( Rocha et al. 2014b)”

Line 504: protein(Arefin et al. 2011 ; Kalambhe et al. 2017) needs spacing “protein (Arefin et al. 2011 ; Kalambhe et al. 2017)”

Line 516: 1 ⁄ 1,000,000 do you mean “10-6”

6. PLOS authors have the option to publish the peer review history of their article (what does this mean?). If published, this will include your full peer review and any attached files.

Reviewer #1: No

Reviewer #2: **Yes: **Nabi Jomehzadeh

Reviewer #3: No

---

## [Author Response · Author response to Decision Letter 0]

18 Nov 2024

Responses to reviewers are attached in separate file.

---

## [Editor Report · Decision Letter 1]

2 Dec 2024

Designing of immunodiagnostic assay using polyclonal antibodies for detection of Enteropathogenic Escherichia coli strains

PONE-D-24-30559R1

Dear Dr. Yinur Mengistu,

We’re pleased to inform you that your manuscript has been judged scientifically suitable for publication and will be formally accepted for publication once it meets all outstanding technical requirements.

Kind regards,

Furqan Kabir

Academic Editor

PLOS ONE
---

## [Editor Report · Acceptance letter]

13 Dec 2024

PONE-D-24-30559R1 

PLOS ONE

Dear Dr. Yinur Mengistu, 

I'm pleased to inform you that your manuscript has been deemed suitable for publication in PLOS ONE. Congratulations! Your manuscript is now being handed over to our production team.

Kind regards, 

on behalf of

Dr. Furqan Kabir 

Academic Editor

PLOS ONE